# Correlation analysis of m$^6$A-modified regulators with immune microenvironment infiltrating cells in lung adenocarcinoma

Wei Ye[1], Tianpeng Huang[2]*

1 Department of Medical Respiratory, Wenzhou Municipal Hospital of Traditional Chinese Medicine, Zhejiang Chinese Medical University Affiliated Wenzhou Hospital of Traditional Chinese Medicine, Wenzhou, China, 2 Department of Clinical Laboratory, Wenzhou Municipal Hospital of Traditional Chinese Medicine, Zhejiang Chinese Medical University Affiliated Wenzhou Hospital of Traditional Chinese Medicine, Wenzhou, China

* huangtianpeng1993@163.com

## Abstract

### Object

Recent studies have demonstrated the epigenetic regulation of immune responses. However, the potential role of N6-methyladenosine methylation (m$^6$A) in the tumor microenvironment (TME) remains unknown.

### Method

In this study, the m$^6$A modification patterns of LUAD samples were comprehensively evaluated by combining TCGA and GEO data, while these modification patterns were systematically linked to the characteristics of immune infiltrating cells in TME. The m$^6$A score was constructed using the principal component analysis algorithm to quantify the m$^6$A modification mode of a single tumor.

### Result

There were three distinct patterns of m$^6$A modification identified. The characteristics of TME cell infiltration in these three patterns were highly consistent with these three immune phenotypes of the tumors, including immune rejection, immune-inflammatory, and immune inert phenotypes. Low m$^6$A scores were characterized by immune activation and poor survival rate. Besides, m$^6$A scores were associated with tumor mutational load (TMB) and were able to increase the ability of TMB to predict immunotherapy. Two immunotherapy cohorts confirmed that the patients with lower m$^6$A scores demonstrated significant therapeutic advantages and clinical benefits. m$^6$A modifications play an important role in the development of TME diversity. Assessing the m$^6$A modification pattern of individual tumors can deepen the understanding as to the characteristics of TME infiltration and guide more effective immunotherapy strategies.

**Data Availability Statement:** All relevant data are within the manuscript and its Supporting Information files.

**Funding:** This work was supported by Zhejiang Wenzhou Science and Technology Program Project (Y20210877).

**Competing interests:** No conflict of competing interest exits in the submission of this manuscript, and manuscript is approved by all authors for publication.

## Introductions

Lung cancer (LC) is one of the common malignant tumors, and lung adenocarcinoma (LUAD) is the most common pathological type [1–4]. Although immunotherapy has gradually become a focus of LUAD treatment, the lack of efficacy indicators and the limited beneficiaries have become the challenges we face [5]. Therefore, it is urgent to find effective solutions. There are more than 100 RNA modifications in organisms [6, 7]. The most common internal modifications of mRNA include N6—adenylate methylation(m$^6$A), N1—adenylate methylation (m1A), and 5—methylcytosine (m5C), etc [8]. These modifications contribute to maintaining mRNA stability and are associated with a variety of diseases such as tumors, neurological diseases, and embryonic development(9). m$^6$A is a methyl insertion on the N atom of adenosine 6, which is considered to be the most significant and abundant form of internal modification in eukaryotic cells. It widely exists in mRNA, lncRNA, and miRNA [9–12]. The methylation modification of m$^6$A has been proved to be reversible, which requires the participation of methyltransferase, demethylase, and methylated reading protein [13]. The methyltransferase such as METTL3, METTL14, WTAP and KIAA1492 form complexes, which can make m$^6$A modifications of mRNA bases and play a catalytic role [13]. Demethylases such as FTO and ALKBH5 play a role in removal. Such methylated reading proteins as YTHDF1, YTHDF2, YTHDF1, YTHDF3, YTHDC1, YTHDC2, and HNRNPA2B1 can recognize the m$^6$A motif, thus affecting the function of m$^6$A [13]. There is mounting evidence showing that the m$^6$A modification gene is closely related to the occurrence and development of tumors, playing a dual role in promoting cancer and inhibiting cancer. Besides, its expression level often affects the pathological evolution of tumors [9].

Tumor tissue includes tumor cells, stromal cells, immune cells, and TME [14]. There is increasing evidence that the diversity of TME plays an important role in the tumor evolution process and immunotherapy, etc [15]. The integrated analysis of TME and m$^6$A modifications has the potential to identify different immune phenotypes of tumors and to improve the guidance and prediction of immunotherapy. A research in this project shows that a comprehensive evaluation of LUAD-m$^6$A modification patterns was performed and a scoring system was established to quantify the m$^6$A modification patterns of patients (The process of this study is described in S1 and S2 Figs).

## Materials and methods

### Data source

The LUAD expression data and complete clinical information were sourced from the cancer genome atlas (TCGA) database and the Gene Expression Omnibus (GEO) database. Transcriptomic expression data (Fragments Per Kilobase of exon model per Million mapped fragments) and the corresponding clinical information data from the TCGA-LUAD dataset, including 535 lung adenocarcinoma tissue samples and 59 paraneoplastic tissue samples, were downloaded from the Genomic Data Commons (GDC https://portal.gdc.cancer.gov/). The transcriptomic expression data was based on the Illumina HiSeq high-throughput sequencing platform. Transcriptomic expression data was annotated according to GENCODE version 29 (https://www.gencodegenes.org/human/). Subsequently, Fragments Per Kilobase of exon model per Million mapped fragments were converted into Transcripts Per Kilobase of exon model per Million mapped reads. A gene expression profile (GSE26939) was downloaded from Gene Expression Omnibus (GEO, http://www.ncbi.nlm.nih.gov/geo/) database by searching for "lung adenocarcinoma" (January 2021). The platform annotation file for

GSE26939 is Agilent-UNC-custom-4X44K. The TCGA-LUAD metagenomic data was downloaded from GDC for copy number variation (CNV) analysis.

## m$^6$A CNV analysis

This study included twenty-three m$^6$A modification related genes, including 8 methylation transferases (METTL3, METTL3, METTL16, WTAP, VIRMA, ZC3H12, RBM15, RBM15B), 13 methylation reading proteins (YTHDC1, YTHDC2, YTHDF1, YTHDF2, YTHDF3 HNRNPC, FMR1, LRPPRC, HNRNPA2B1, IGFBP1, IGFBP2, IGFBP3, RBMX), and 2 demethylases (FTO, ALKBH5). First of all, the copy number of the m$^6$A regulators was extracted at TCGA-LUAD using PERL software, and the histogram was constructed visually using R software. To explore the relationship between the copy number of 23 m$^6$A regulators and chromosomes, the RCircos package was used to plot the variation in copy number of 23 m$^6$A regulators in 23 pairs of chromosomes. The Wilcox test was performed to compare the differential expression of m$^6$A regulators in TCGA-LUAD using the limma package. Limma package provides a very comprehensive solution to microarrays analysis and RNA-Seq differential analysis [16]. Waterfall plots were drawn with the maftools package to demonstrate the mutation rate of m$^6$A regulators in LUAD. The m$^6$A regulators with higher mutation rates were selected to divide the samples into wild and mutant groups for comparing the relationship between gene and expression. P-value < 0.05 was treated as statistically significant, and the box plot was drawn using the ggpubr package.

## m$^6$A regulator analysis

The TCGA-LUAD datasets and GSE26939 datasets were subjected to intersection taking, data merging, data correction, and the removal of normal samples for further analysis. Prognosis-related m$^6$A regulators were selected using the Univariate Cox regression model and the survival package with P-value<0.05 as the cut-off condition. The relationships between prognosis-associated m$^6$A regulators were further demonstrated in the form of network diagrams using the igraph package.

**m$^6$A cluster.** To further understand the value of m$^6$A regulators, the samples were clustered by the ConsensusClusterPlus package according to the expression of m$^6$A regulators. All samples were set into k [2–9] groups, and after sequential cycling, the most appropriate clustering typing of m$^6$A regulators was obtained according to 3 conditions. The first one is tight intra-typical associations and non-tight inter-typical associations. The second one is that there are not too few samples within each cluster. The last one is an insignificant increase in the area of the cumulative distribution curve. Based on the correlation between the m$^6$A cluster and survival status, the cut-off points for each data set subgroup were determined using the survminer package, while repeated tests were performed for all possible cut-off points to find the maximum rank statistic. Then, the patients were divided into high expression group and low expression group based on the maximum selected log-rank statistic. The survival curves for predictive analysis were generated using the Kaplan-Meier method and the survival package, while the log-rank test was performed to determine the significance of differences, with P-value < 0.05 treated as statistically significant.

**GSVA analysis and ssGSEA analysis.** To investigate the biological functions among m$^6$A regulators, gene set variation analysis (GSVA) was performed using the GSVA package. GSVA is a non-parametric and unsupervised method mainly used to estimate the changes in pathways and the biological process activity of samples in experimental datasets [17]. Gene sets were downloaded from the MSigDB database (http://www.gsea-msigdb.org/gsea/msigdb) for c5.go.v7.4.symbols, and p-values were adjusted according to the false discovery rate (FDR),

with P-value < 0.05 as the cut-off criterion. Heatmaps were plotted using the pheatmap package.

The relative abundance of immune infiltrating cells (immune score) in each sample was quantified using gene enrichment score (NES) and a single sample GSEA (ssGSEA) [18]. The gene set for each TME-infiltrating immune cell type was derived from the study of Charoentong, including activated CD8 T cells, activated dendritic cells, macrophages, natural killer T cells, and regulatory T cells, etc [19]. The correlation between m$^6$A typing and immune scoring was further explored using the limma package, with P- value < 0.05 treated as statistically significant, and box plots were obtained using the ggpubr package.

**Differential analysis.** Bayesian statistics of the R software limma package were applied to determine the differential genes (DEGs) between the two groupings of m$^6$A, p-values were adjusted according to the false discovery rate (FDR), and the P-values smaller than 0.001 were taken as the screening criteria. The core genes were obtained by taking the intersection of DEGs between different genotypes using the VennDiagram package. Furthermore, GO enrichment analysis and KEGG enrichment analysis were performed through Matescape database (http://metascape.org) to explore the potential biological functions and biological pathways, and potential biological functions and pathways were selected with P -value < 0.05.

## Gene cluster

Univariate Cox regression models analysis of DEGs was conducted using survival package to screen out prognosis-related m$^6$A phenotype modifying genes with P-value <0.05. The samples were clustered according to the expression of prognosis-related m$^6$A phenotype modifying genes using the ConsensusClusterPlus package, so as to identify gene clusters for the next step of analysis(the same as step-m$^6$A cluster).

Firstly, the survival package of R software was applied to perform survival analysis to help assess the prognostic value of gene cluster and was divided into high and low expression groups. Besides, survival curves were plotted using the Kaplan-Meier method, and the log-rank test was performed to assess statistical significance, with a P-value < 0.05 treated as statistically significant (the same as step-m$^6$A cluster). After the collection of clinical information (i.e., clinical-stage, T stage, N stage, M stage, age, and sex), the pheatmap package was used to draw a heat map showing the correlation between gene cluster, m$^6$A cluster, and clinical features.

## m$^6$A score

In order to quantify the m$^6$A modification pattern of LUAD, a scoring system was established by principal component analysis (PCA) to evaluate the characteristics of LUAD-m$^6$A modification. PCA analysis can be effective in identifying the most dominant elements and structures in the data, removing noise and redundancy, reducing the dimensionality of the original complex data, and revealing the simple structure hidden behind the complex data [20]. Principal component analysis was conducted to construct the scoring system:

m6A score = (PCli * PC2i)

i denotes the expression of the m$^6$A gene

m$^6$A score divided into high and low m$^6$A score groups for further analysis (the same as step-m$^6$A cluster). Firstly, survival analysis was performed using the survival package to help assess the prognostic value of the m$^6$A score group, and survival curves were plotted using the Kaplan-Meier method with Lonkrank test to assess statistical significance (the same as step-m$^6$A cluster), with a smaller P-value than 0.05 considered statistically significant. Further clinical information (pathological staging, survival status) was incorporated, and the survival

curves in different pathological staging were plotted using the same method. Then, the correlation between m6A typing, genotyping, m6A score group, and survival status was explored using ggalluvia package for mulberry plots. Furthermore, the differences in m6A scores in different pathological stages and survival status were calculated by counting the differences in m6A scores, percentage plots were plotted using the plyr package, and Box-line plots were plotted using ggpubr package. The differences in m6A scores between different m6A staging and gene were compared using the limma package. P-value < 0.05 was treated as statistically significant.

## Correlation between tumor mutation burden and m6A score

The tumor mutational burden (TMB) was defined as the total number of somatic gene coding errors, base substitutions, gene insertion, or the deletion errors as detected per million bases. Firstly, the TMB in each sample was extracted by PERL software. The box line plots and correlation graphs showing the relationship between TMB and m6A score groups were drawn using the ggpubr package. Then, survival analysis was performed using the survival package. All samples were divided into high group(>median value) or low expression group(<median value) according to the expression of TMB. The Kaplan-Meier method was adopted to draw survival curve, and the log-rank test was performed to assess statistical significance. P-value<0.05 was considered statistically significant. The same method was used to plot the survival curves for TMB combined with the m6A score. Finally, waterfall plots were drawn using the maftools package to demonstrate the mutation rates between the high and low m6A score groups.

## Analysis of immune˚checkpoint

Firstly, the correlation between immune scoring and m6A score was compared using the corrplot package of R software. Then, the samples in the m6A score group and clinical information samples (survival status) were intersected while the data was combined using the R software. The TCIA database (https://tcia.at/home) stores high-throughput sequencing data and immunogenomic analysis results for more than 20 cancers, including the gene expression of relevant tumors, immune infiltrating cell composition, neoantigens, carcinoembryonic antigens, and others. Immunotherapy scoring files were obtained through the TCIA website, and violin plots were drawn using the ggpubr package to compare the relationship between high and low m6A scoring groups and immune checkpoint inhibitors. In addition, the expression of PD-L1 in each sample was further extracted and the correlation between the m6A score and PD-L1s was analyzed using the limma package. A P-value smaller than 0.05 was considered statistically significant.

## Statistics analysis

The correlation coefficients between TME infiltrating immune cells and m6A regulator expression were calculated by means of Spearman and distance correlation analysis. The one-way analysis of variance and Kruskal-Wallis test were conducted to compare the differences among three or more groups. The survival curves for analysis were generated using the Kaplan-Meier method, and the log-rank test was performed to determine the significance of differences. Univariate Cox regression models were adopted to calculate hazard ratios (HR) for m6A regulators and m6A-associated genes. All data processing was carried out using R (version 4.0.3) and PERL software (version 5.10.0).

## Results

### Epigenetic analysis of m⁶A in lung adenocarcinoma

This study involved 23 m⁶A genes, including 8 methylation transferases, 13 methylation reading proteins, and 2 demethylation enzymes. As shown in Fig 1A, there are 115 samples mutated in 561 samples, with an incidence of 20.5%. A higher mutation rate occurred in ZC3H13 (mutation rate of 3%), while no mutation rate occurred in METTL3 or VIRMA. Copy number variation analysis revealed that the significant increase in copy number occurred in YTHDF1, VIRMA, FMR1, RMR1, METTL3, HNRNPC, RBMX, LRPPRC, and HNRNPA2B1, while extensive copy number deletions were present in YTHDF2, YTHDC1, YTHDC2, RBM15, and METLL14 (Fig 1B). The locations of CNV alterations in m⁶A

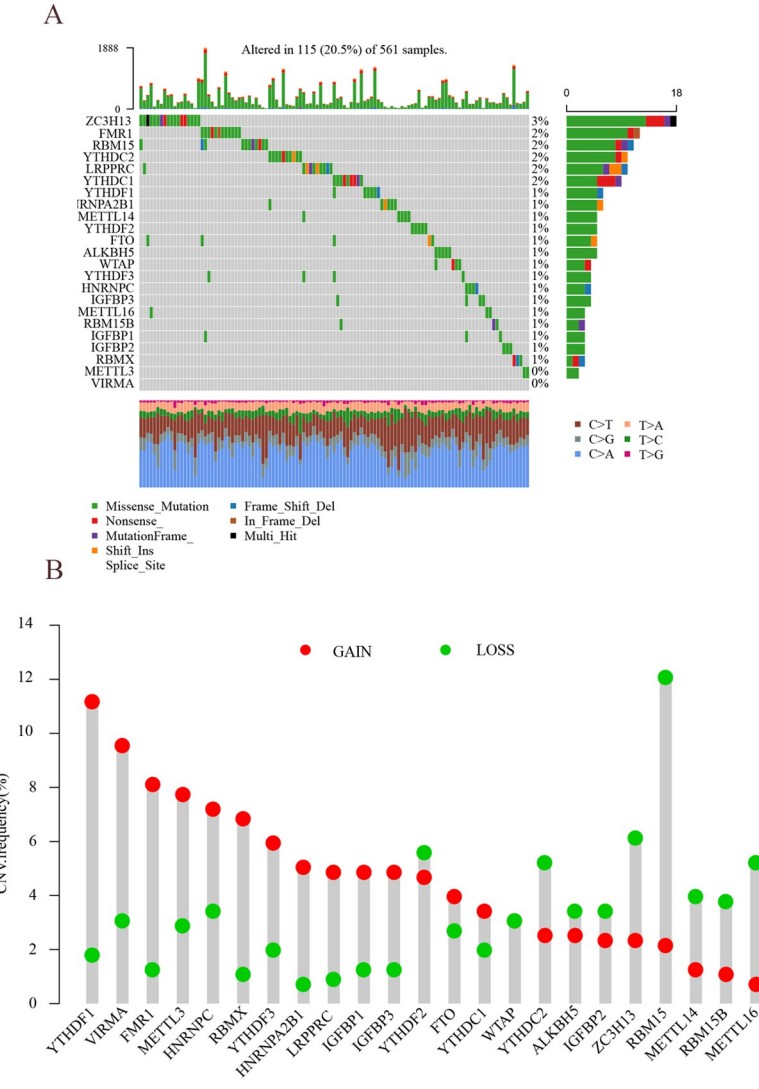

**Fig 1. Epigenetic results of m⁶A in lung adenocarcinoma.** (A) m⁶A waterfall plot. The right vertical coordinate represents m⁶A regulators, and the left vertical coordinate represents the mutation rate of m⁶A regulators in LUAD. (B) m⁶A copy number variation frequency. The horizontal coordinate represents m⁶A regulators, the vertical coordinate represents CNV mutation rate, red circles indicate gene amplification, and green circles indicate gene deletion.

regulators on chromosomes are shown in Fig 2A. Both LUAD tissues and adjacent non-cancerous tissues could be identified according to the CNV alterations in chromosomes. To further investigate the relationship between regulators and epigenetics, the expression levels of $m^6A$ regulators were further analyzed, as shown in Fig 2B. Most $m^6A$ regulators such as METTL3, METTL14, METTL16, WTAP, VIRMA, ZC3H12, RBM15, and RBM15B were

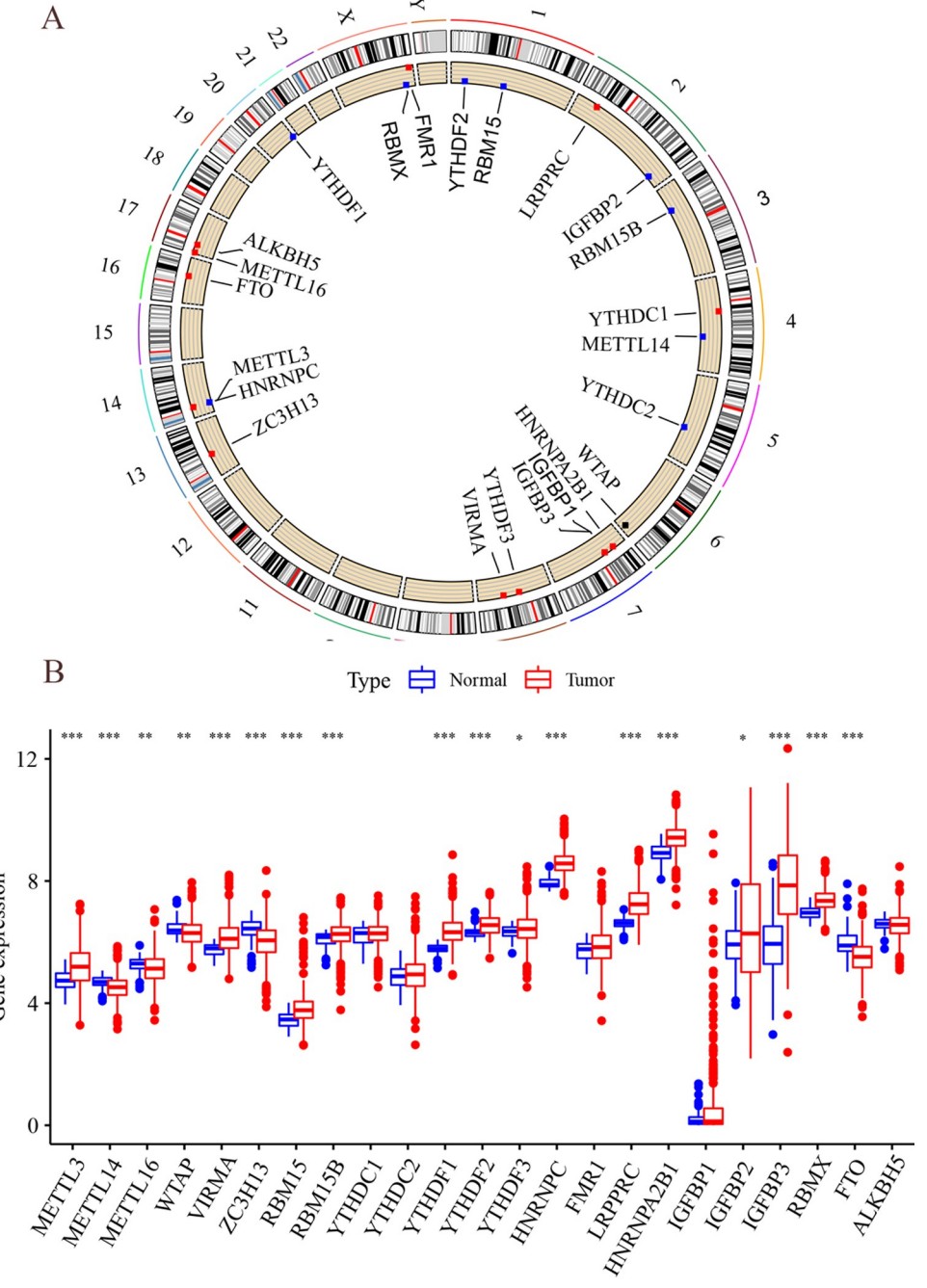

**Fig 2. Copy number of $m^6A$ in chromosomes and analysis of differences.** (A) Copy number circle plot. Distribution of $m^6A$ regulators in 22 pairs of autosomes and 1 pair of sex chromosomes, red circles indicate copy number increase, and blue circles indicate copy number decrease. (B) TCGA-$m^6A$ differential analysis. Horizontal coordinates indicate $m^6A$ regulators, vertical coordinates indicate gene expression, * P < 0.05, * * P < 0.01, and * * * P < 0.001.

differentially expressed in LUAD tissues and adjacent non-cancerous tissues to a significant extent (P<0.05). Copy number variation may lead to the altered expression levels of m$^6$A regulators, and there were highly specific epigenetic alterations in m$^6$A regulators in tumors and adjacent non-cancerous tissues.

## m$^6$A regulators

The GEO data with survival time and clinical information was introduced into the study. Univariate regression models revealed that 14 m$^6$A modified regulators (ALKBH5, FMR1, HNRNPA2B1, HNRNPC, IGFBP2, LRPPRC, IGFBP3, METTL3, RBM15, VIRMA, YTHDC1, YTHDC2, YTHDF1, YTHDF2) had a high prognostic value (S1 Table) (P<0.05). As revealed by the network diagram of m$^6$A gene interaction relationship, the m$^6$A modified regulators in the same category had a significant correlation, as did the m$^6$A modified regulators in different categories (Fig 3A). For example, HNRNPCY could inhibit the expression of YTHDC2, FTO, METTL16, and HNRNPCY was co-expressed with LRPPRC, YTHDF3, WTAP, and VIRMA. While HNRNPCY and RBM15 had a high mutation frequency, for which the samples were divided into wild and mutant groups (Fig 3B), with the results suggesting that LRPPRC was significantly up-regulated in the mutant group compared with the wild group (P<0.05). The samples were clustered according to the expression of m$^6$A regulators and then divided into cluster A, cluster B, and cluster C (S2 Table). Cluster A contained 219 samples, cluster B contained 208 samples and cluster C contained 202 samples.

## Functional analysis of m$^6$A cluster

To further explore the potential biological functions among m$^6$A fractions, GSVA analysis was performed (Fig 4A–4C). Cluster A is mainly related to specific immune response and activation, such as the differentiation of helper T lymphocytes and the regulation of related signaling pathways. Cluster B is mainly related to the intrinsic immune response, such as the regulation of TOLL-like receptor signaling pathway and NF-κb transcription factor activity. Subsequently, the relationship between m$^6$A cluster and immune infiltrating cells was further analyzed (Fig 5). It was found out that cluster A had abundant immune infiltrating cells, including CD4$^+$ T lymphocytes, CD8$^+$ T lymphocytes, and regulatory T lymphocytes (P<0.05), all of which were jointly involved in the specific immune response. While cluster B included mast cells, monocytes, γδ T cells, dendritic cells, etc. (P<0.05), as involved in the non-specific immune response. m$^6$A cluster exists with distinctly different cellular infiltration characteristics of the tumor microenvironment. Cluster A is immunoinflammatory, cluster B is immune rejection, and cluster C is immune desert.

## m$^6$A phenotype-related genes

The limma package was applied to perform differential analysis among the m$^6$A cluster (Fig 6). The results showed that there were 1654 differential genes between cluster A-B, 3592 differential genes between cluster A-C, and 5194 differential genes between cluster B-C. The differential genes among m$^6$A typing were taken as common intersection, i.e. 176 differential intersection genes, which were considered as the core genes. The GO analysis and KEGG analysis were performed through the matescape online database, while significant biological functions and pathways were screened at P<0.05. The genes for GO analysis were enriched in the regulatory functions of cytokines and maintenance of cellular homeostasis (Fig 7A). The KEGG functional analysis suggested that they were mainly enriched in the MAPK signaling pathway (Fig 7B), indicating that these core genes were significantly associated with immune regulation.

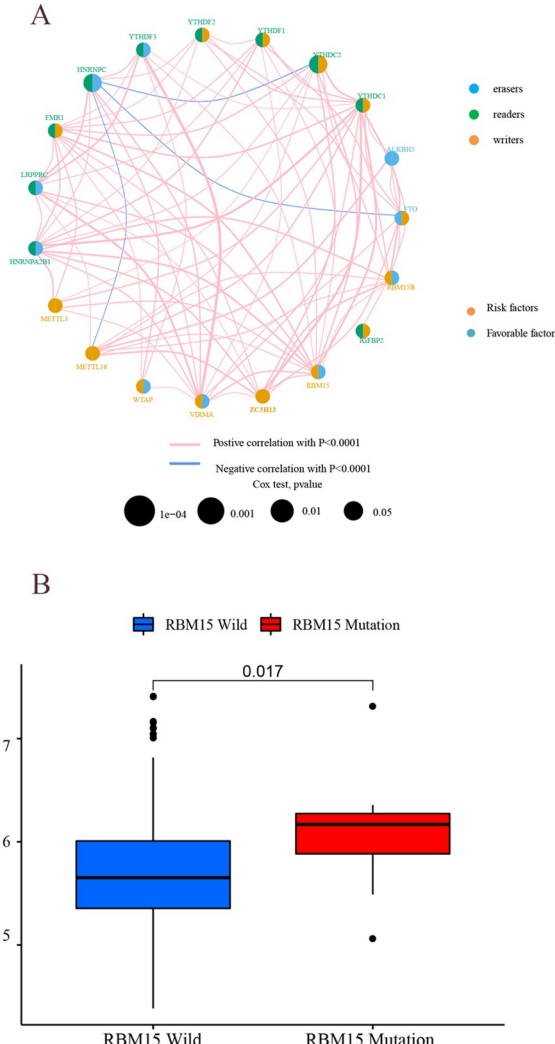

**Fig 3. Analysis of m⁶A regulators.** (A) m⁶Aprognostic network diagram. In the circle corresponding to each m⁶A modified gene, the yellow part of the left half-circle indicates risk genes, the blue part indicates protection genes, the blue part of the right half-circle indicates demethylase, the green part indicates methylated reading protein, the yellow part indicates methyltransferase, the pink linkage between genes indicates synergism, and the blue linkage indicates repression. (B) Mutation and expression correlation analysis. The horizontal coordinates indicate the RBM15 wild group and mutant group, respectively, while the vertical coordinates indicate the LRPPPC expression.

## Gene cluster

There was no significant difference in survival for the m⁶A cluster (Fig 8A). According to the expression of core genes (S3 Table), the samples were classified into A, B, and C by cluster analysis (S4 Table). Median survival was significantly different between the 3 groups (P<0.001): cluster B > cluster A > cluster C (Fig 8B). The distribution structure of m⁶A cluster A (immunoinflammatory type) in genotyping is: cluster B > cluster A >cluster C (Fig 9). The variability of genotypic survival may be associated with the m⁶A cluster (immunoinflammatory type). The possible reason for this is that immune-inflammatory type tumors have a large number of immune infiltrating cells in the microenvironment, which are sensitive to immune checkpoint inhibitors and have a better prognosis. Furthermore, it was observed that gene cluster B was mainly concentrated in stage I-II (Fig 10), and cluster B had a longer median survival, which may also be related to the fact that the patient was in the early stage of the tumor.

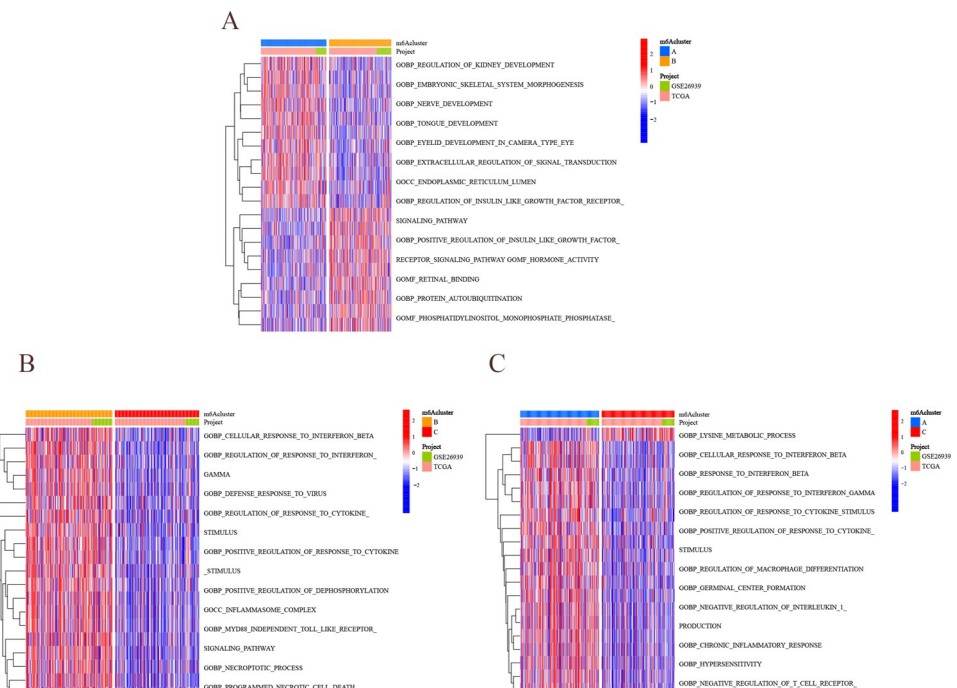

**Fig 4. Results of m⁶A typing analysis.** (A) The first row indicates m⁶A cluster (blue represents cluster A, and yellow represents cluster B), the second row indicates the dataset (green represents GEO dataset, and pink represents TCGA dataset), and the third column indicates the functional enrichment results (red represents positive correlation, and blue represents negative correlation). (B) The first row indicates m⁶A cluster (red represents cluster B, and yellow represents cluster C), the second row indicates the dataset (green shows GEO dataset, and pink shows TCGA dataset), and the third row indicates the function enrichment result (red indicates positive correlation, and blue indicates negative correlation). (C) The first row indicates the m⁶A cluster (blue is cluster A, and red is cluster C), the second row indicates the dataset (green is the GEO database, and pink is the TCGA database), and the third row indicates the functional enrichment results (red indicates positive correlation, and blue indicates negative correlation).

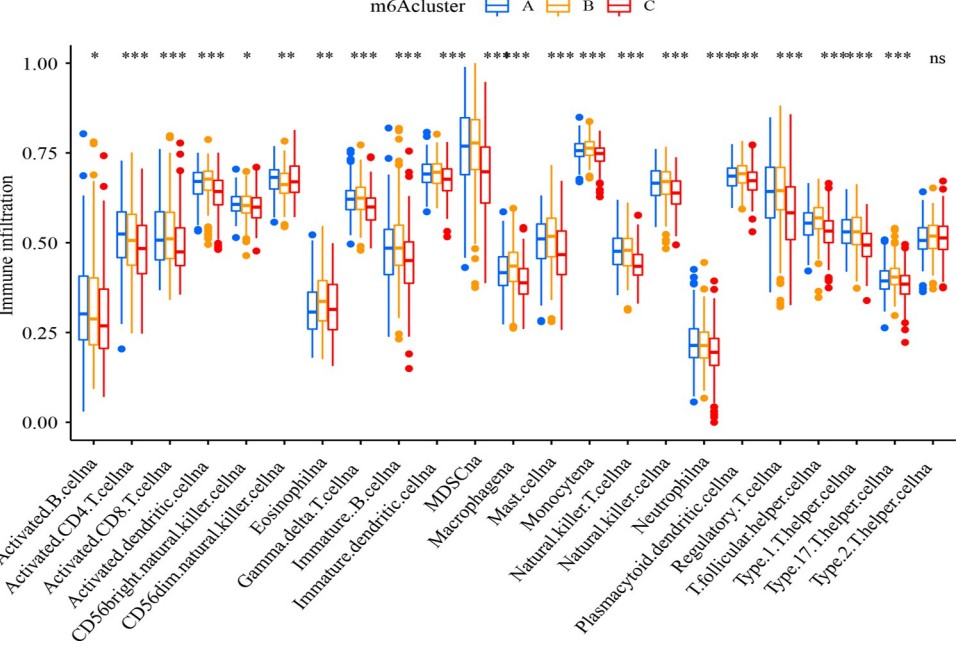

**Fig 5. Immune cell differential analysis (ssGSEA).** Horizontal coordinates indicate immune infiltrating cells, vertical coordinates indicate immune infiltration abundance, blue boxes indicate cluster A, yellow boxes indicate cluster B, and red boxes indicate cluster C. * P<0.05, * *P<0.01, and * * * P<0.001.

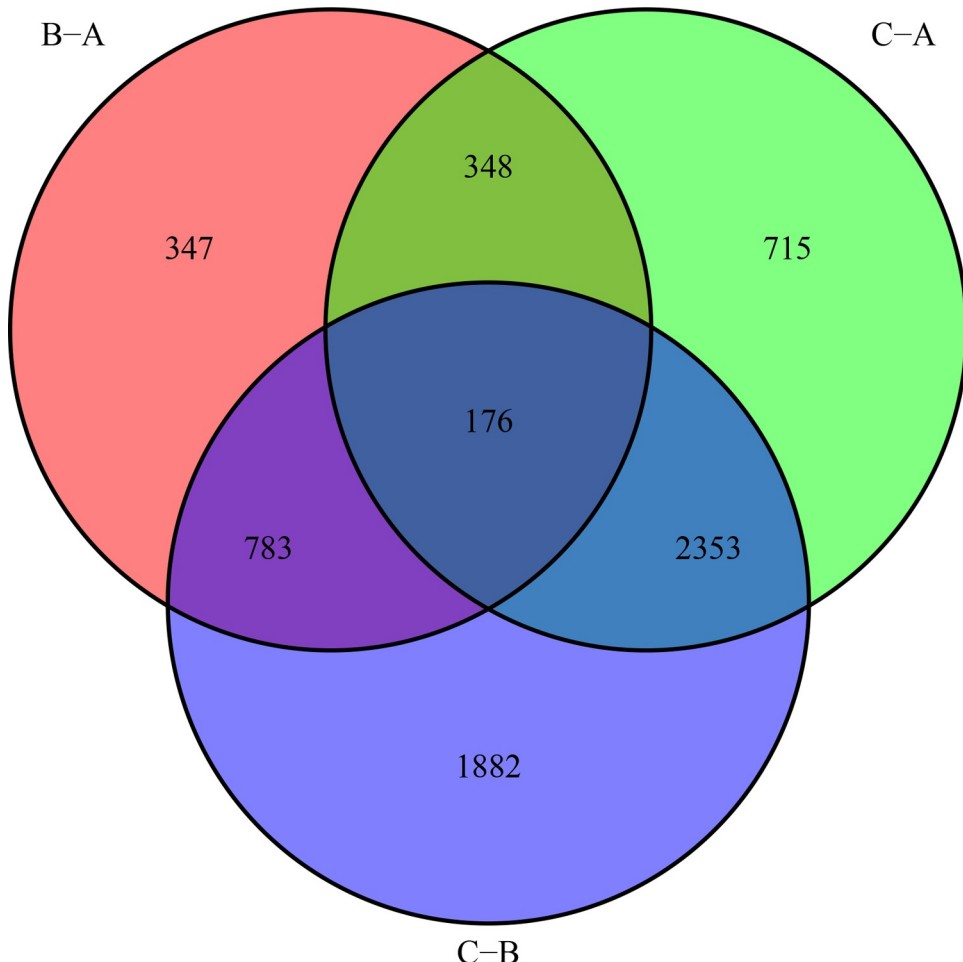

**Fig 6. m⁶A cluster differential genes.** The red part indicates B-A differential genes, the green part indicates C-A differential genes, and the purple part indicates C-B differential genes.

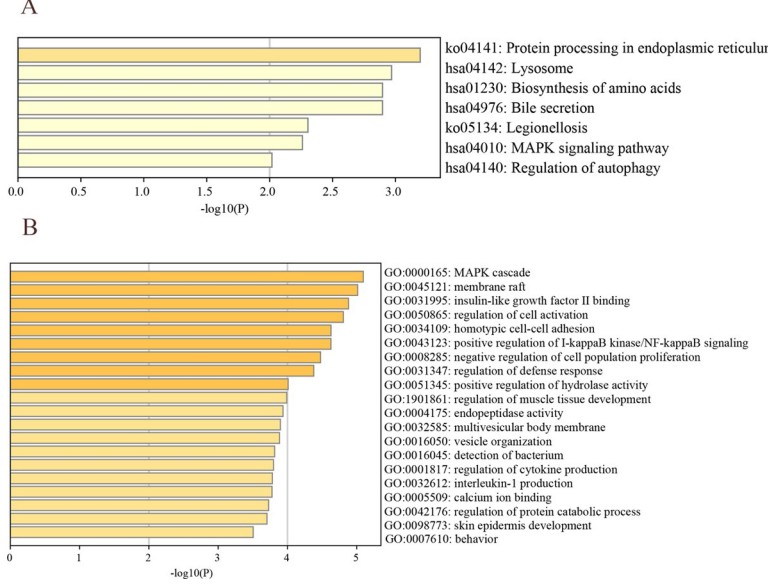

**Fig 7. Functional analysis of differential genes among m⁶A cluster.** (A) GO functional analysis of core genes. (B) KEGG functional analysis of core genes.

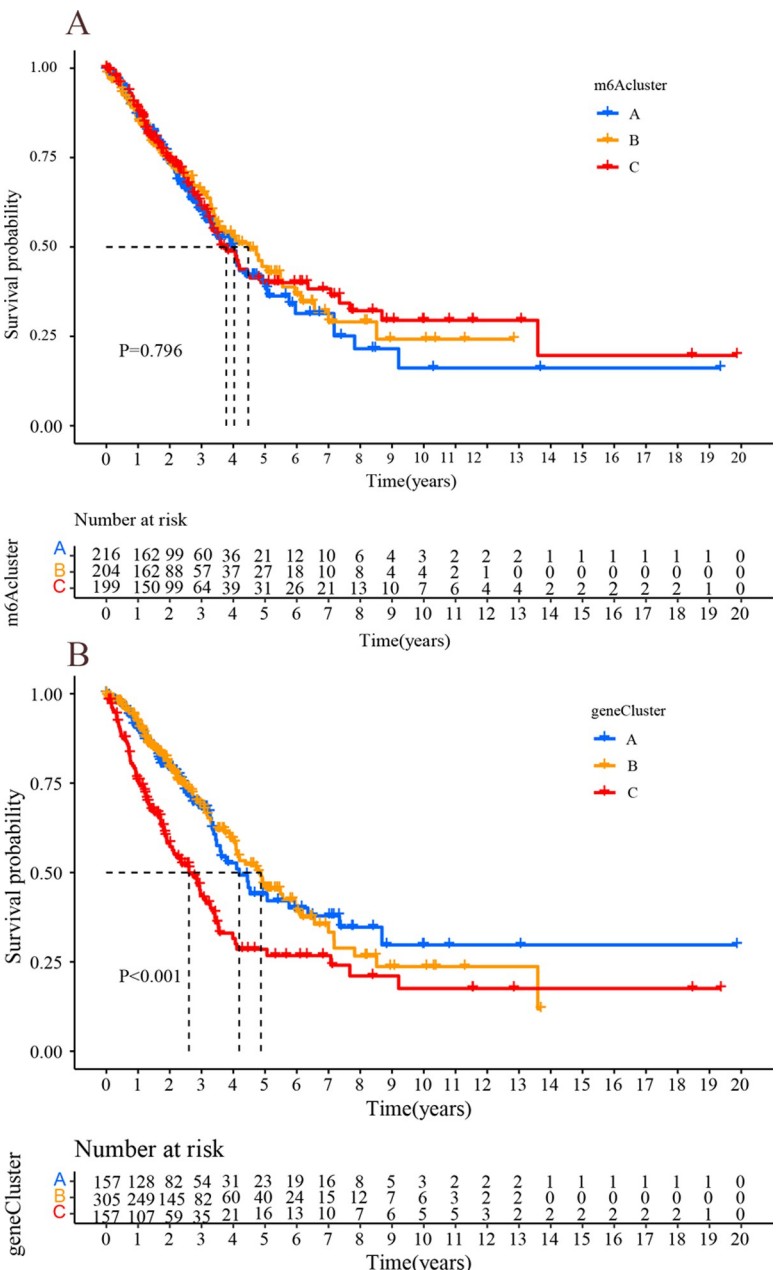

**Fig 8. Survival analysis results.** (A) m⁶A cluster survival analysis: horizontal coordinates indicate survival time, vertical coordinates indicate survival rate, the blue line indicates cluster A, the yellow line indicates cluster B, and the red line indicates cluster C. (B) genecluster survival analysis:horizontal coordinates indicate survival time, vertical coordinates indicate survival rate, the blue line indicates gene cluster A, the yellow line indicates gene cluster B, and the red gene line indicates cluster C.

## m⁶A score

The m⁶A score was used to quantify the m⁶A modification pattern, and the samples were divided into high m⁶A score group and low m⁶A score group by survminer package (S5 Table). The low m⁶A score group had a longer survival (Fig 11A, P<0.001) with 70% survival and 30% mortality in the low m⁶A score group compared to 52% survival and 48% mortality

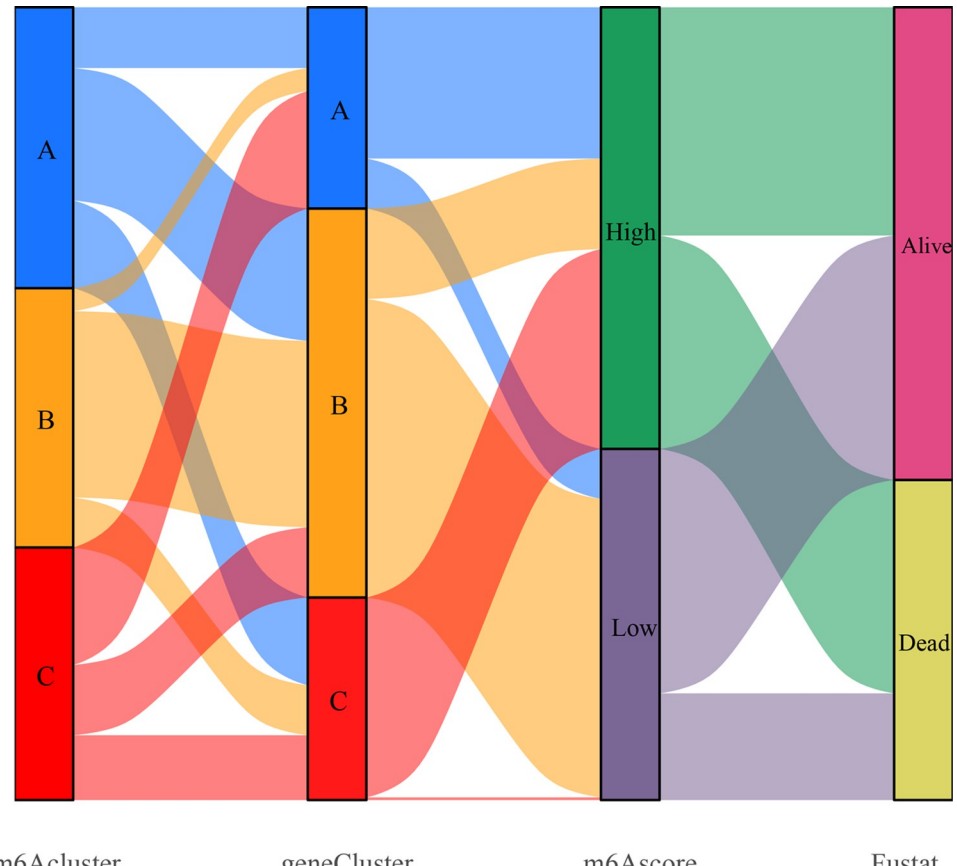

m6Acluster geneCluster m6Ascore Fustat

**Fig 9. Genotyping m$^6$A sang-froid.** The first row indicates m$^6$A cluster A, B, and C, the second row indicates gene cluster A, B, and C, the third row indicates high and low m$^6$A scores, and the fourth row indicates survival status (alive, dead), with correlations indicated by connecting lines between different rows.

in the high m$^6$A score group (Fig 12A and 12B). In further analysis, there was a significant difference in survival in patients with stage I-II: the lower scoring group had a better prognosis (P<0.001); while there was no significant difference in survival in stage III-IV (Fig 11B and 11C). m$^6$A scores were higher the later the stage was staged (P<0.001), with 85% of stage I-II patients in the low m$^6$A score and 15% of stage III-IV patients in the low m$^6$A score (Fig 12C and 12D). The m$^6$A score may be an independent prognostic factor for patients with early LUAD. In addition, there was significant variability (P<0.001) between the scores of m$^6$A subtype B and type A and C, respectively, further demonstrating that the m$^6$A score can be used to assess m$^6$A subgroup (Fig 11D).

## m$^6$A score and TMB

The mutation rates of the high m$^6$A score group and low m$^6$A score group were analyzed separately using the maftools package (Fig 13A and 13B). The high m$^6$A score group (95.62% mutation rate) exhibited a wider range of mutations than the low m$^6$A score group (80% mutation rate). Further analysis revealed a positive correlation between m$^6$A scores and TMB, with high m$^6$A scores showing higher TMB (Fig 14A), which is consistent with previous findings. Then, the high-scoring group did not show higher survival, which is probably because most of the patients in the high-scoring group were in the advanced tumor stage (Fig 14B). The patients

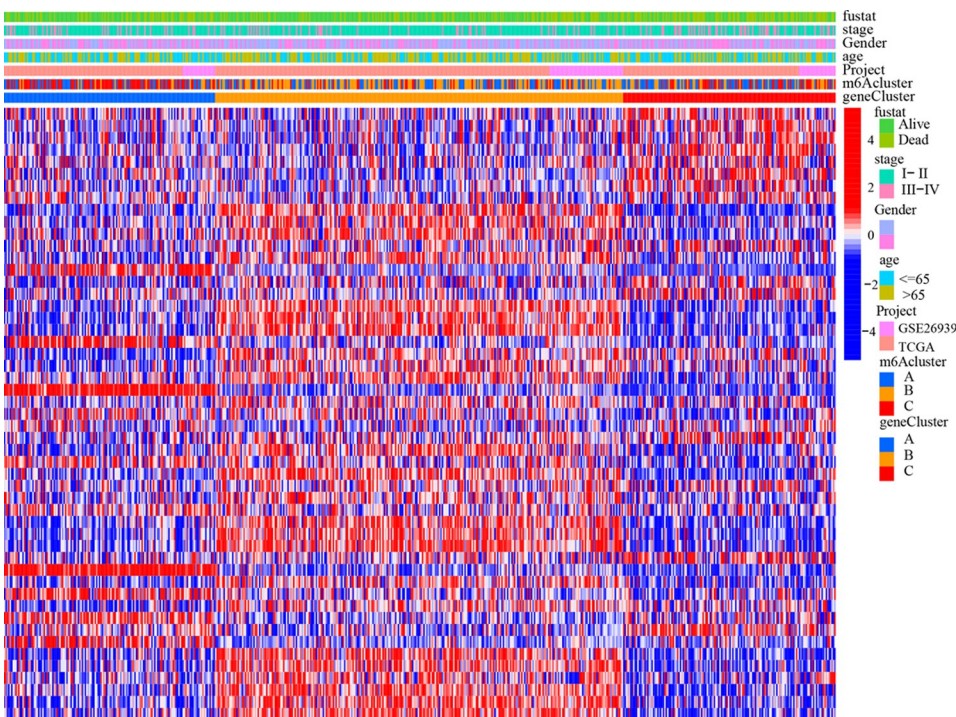

**Fig 10. Gene cluster heatmap different clinical features.** m$^6$A cluster, clinical characteristics and distribution of m$^6$A modified genes in gene cluster.

with high TMB status had a durable clinical response to anti-PD-1/PD-L1 immunotherapy [8]. Therefore, the above results indirectly demonstrate that tumor m$^6$A-modified genes may be a key factor mediating the clinical efficacy of anti-PD-1/PD-L1 immunotherapy. The survival of LUAD patients was not significantly different between high and low TMB (Fig 14C), but it was further found out that TMB combined with m$^6$A score predicted the survival of LUAD patients differently (Fig 14D): high TMB + low m$^6$A score > low TMB + low m$^6$A score > high TMB + high m$^6$A score > low TMB + low m$^6$A score (P<0.001). It can be speculated that the combined m$^6$A score can improve the sensitivity of TMB to predict immunotherapy in LUAD.

## Analysis of immune°checkpoint molecules

The results of correlation analysis showed a significant negative correlation between immune infiltrating cells and m$^6$A score (Fig 15A), i.e., the lower the m$^6$A score, the more immune infiltrating cells, which is similar to immune-inflammatory tumors, and the better the prognosis, indirectly demonstrating that m$^6$A score can be used to distinguish tumor immune phenotypes. LUAD patients in the low m$^6$A score group had higher expression of PD-L1 (Fig 15B), suggesting a better response to PD-L1/PD-1 immunotherapy. m$^6$A low score group received PD-L1, CTLA-4, and combination therapy with better results than the high m$^6$A score group (Fig 16A–16D, P<0.001). The above results suggest that the m$^6$A score is a potential and reliable biological indicator for prognosis and the clinical efficacy assessment of immunotherapy.

## Discussion

Increasingly, m$^6$A regulators are being studied to play an important role in multiple aspects of inflammation, tumor, and immunity [21]. Current tumor studies have focused on the role of

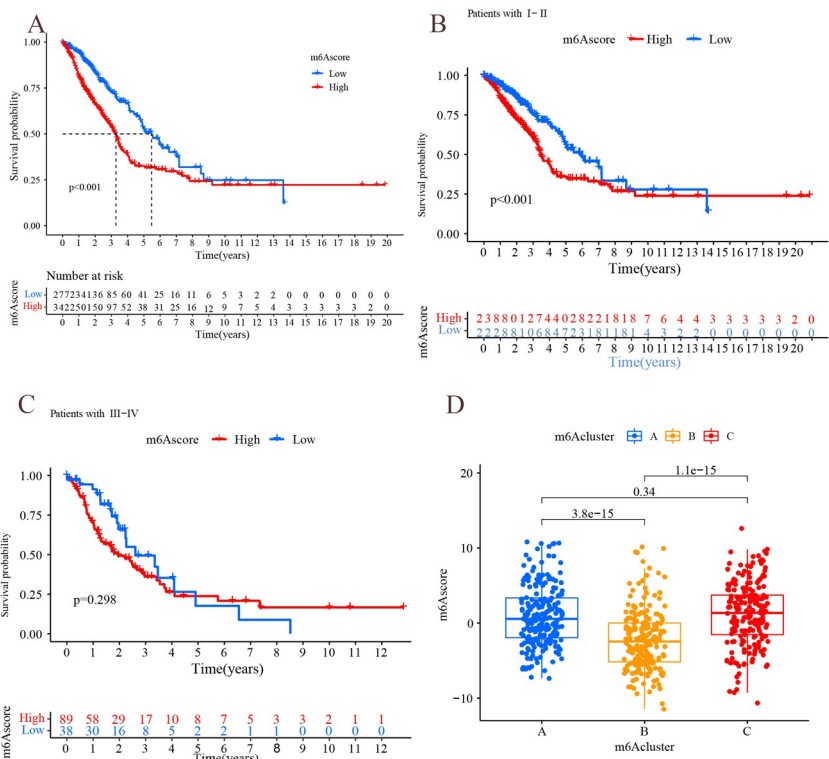

**Fig 11. Results of m⁶A score analysis.** (A) Survival analysis in m⁶A score subgroup: Horizontal coordinates indicate survival time, vertical coordinates indicate survival rate, red line indicates high m⁶A score group, and blue line indicates low m⁶A score group. (B) Survival analysis in m⁶A scoring groupings(patients with stage I-II): Horizontal coordinates indicate survival status, and vertical coordinates indicate m⁶A score. (C) Survival analysis in m⁶A scoring groupings(patients with stage II-III): Horizontal coordinates indicate m⁶A score, vertical coordinates indicate patient proportion, blue square part indicates survival status, and red square part indicates death status.

individual genes, but the role of multiple m⁶A regulators in the tumor microenvironment has been less studied. Exploring the relationship between m⁶A regulators and tumor microenvironment can not only help us understand the tumor microenvironment and tumor evolution, but also guide the development of immunotherapy protocols more effectively [22].

There are 23 m⁶A modifier genes included in this study, revealing three different m⁶A typings of LUAD. These three different patterns of m⁶A typing have distinctly different immune infiltrating cell characteristics. Cluster A is dominated by adaptive immune activation and corresponds to the immune-inflammatory type. Cluster B is dominated by intrinsic immunity and stromal activation and corresponds to the immune rejection type. Cluster C lacks immune infiltration and antigen presentation and corresponds to the immune desert type. Immunoinflammatory tumors are those in which there are more infiltrating lymphocytes in the tumor microenvironment. The immunorejection tumors are those in which immune cells are embedded in the tumor mesenchyme and appear to penetrate the mesenchyme. In fact, however, they maybe confined to the tumor envelope. The immunodesert tumors are those without immune infiltrating lymphocytes [23, 24]. A further functional analysis also confirms the above findings. Cluster A of the functions performed by m⁶A cluster is mainly enriched in T lymphocyte regulation and activation related. Cluster B of the biological functions performed by m⁶A cluster is mainly focused on the intrinsic immune response. In summary, it provides a new direction for the development of immunophenotypic typing under different m⁶A modifications in LUAD.

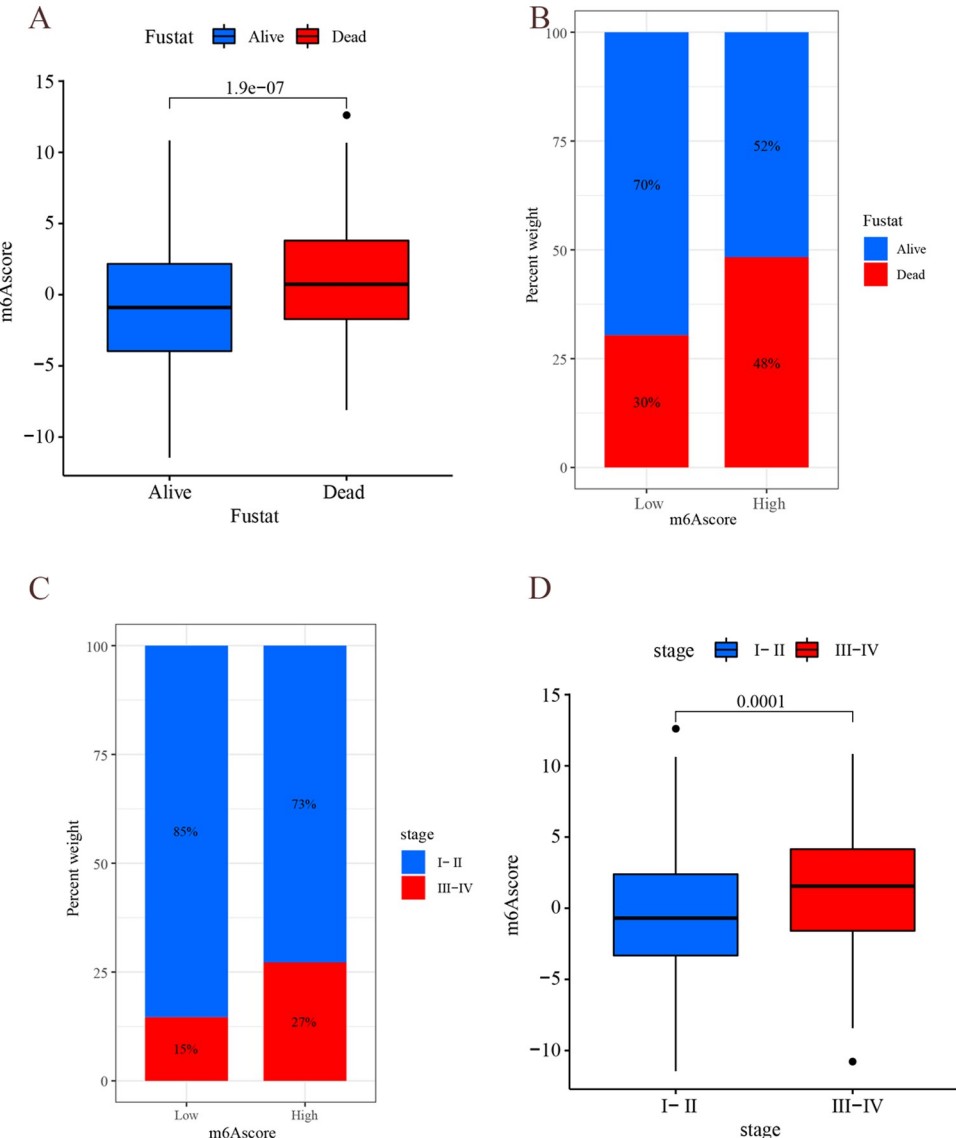

**Fig 12. Results of m⁶A score analysis.** (A) The horizontal coordinate indicates the survival state, and the vertical coordinate indicates m⁶A score. (B) The horizontal coordinate indicates m⁶A score group(high m⁶A score group or low m⁶A score group), and the vertical coordinate indicates survival rate (the red line indicates "Alive", the blue line indicates "Dead"). (C) The horizontal coordinate indicates m⁶A score type(the red line indicates stage I-II, the blue line indicates stage III-IV), and the vertical coordinate indicates survival rate. (D) The horizontal coordinate indicates the stage, and the vertical coordinate indicates m⁶A score.

In addition, the DEGs of m⁶A cluster were considered as the core genes of m⁶A and were shown to be significantly associated with tumor-associated biological pathways. m⁶A cluster did not show significant differences in survival among m⁶A type. With the core genes of m⁶A further used, three gene clusters were identified, possessing characteristic gene clusters with significant differences in survival. In addition, the survival of genotypes was correlated with the immunoinflammatory phenotype, and it is possible that m⁶A modified regulators play a role in shaping the tumor microenvironment. To further quantify m⁶A modifications, a set of m⁶A scores were established that showed a significant negative correlation with immune infiltrating cells. Besides, the lower the m⁶A score value, the more significant the immune-

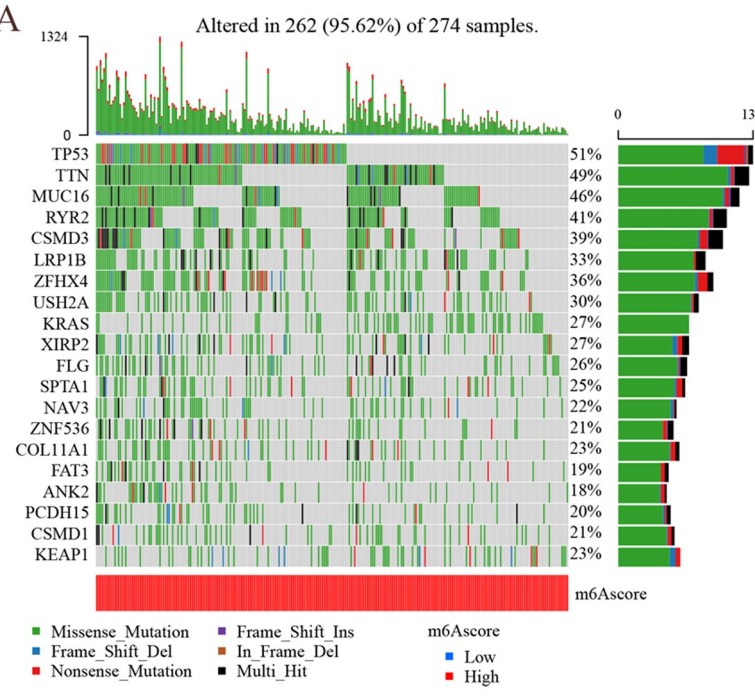

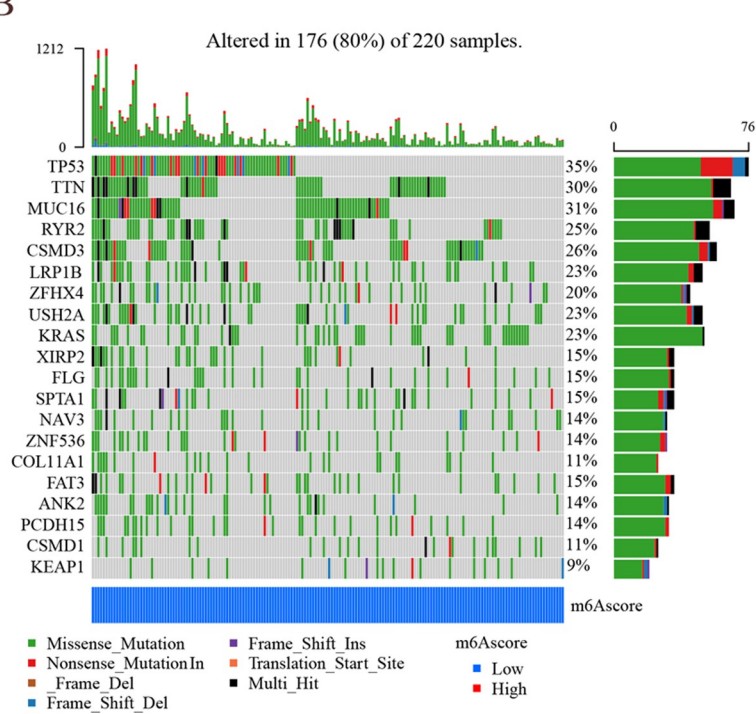

**Fig 13. Mutation waterfall plot for high and low m⁶A score groups.** (A) The mutation rates of high m⁶A score groups. The left vertical coordinate indicates m⁶A-related modified genes, and the right vertical coordinate indicates gene mutation rate in LUAD. (B) The mutation rates of low m⁶A score groups. The left vertical coordinate indicates m⁶A-related modified genes, and the right vertical coordinate indicates gene mutation rate in LUAD.

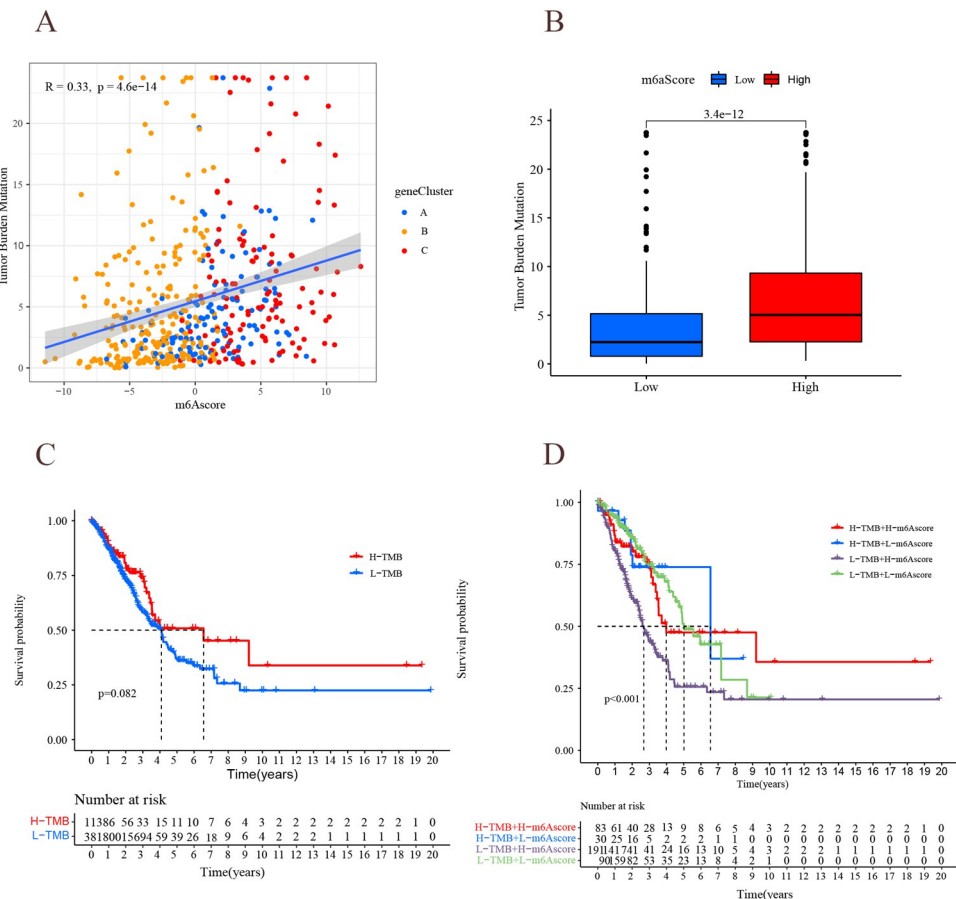

**Fig 14. Relationship between mA6 scoring and TMB.** (A) m⁶A score and tumor mutation load. Horizontal coordinate indicates m⁶A score, vertical coordinate indicates TMB, and scatter indicates genecluster. Blue: genecluster A; yellow: genecluster B; red: genecluster C. (B) genecluster-TMB analysis. Horizontal coordinates indicate high and low m⁶A scores, and vertical coordinates indicate TMB. (C) TMB survival analysis. Horizontal coordinates indicate survival time, and vertical coordinates indicate survival rate. Red line: high TMB group; blue line: low TMB group. (D) TMB combined m⁶A score survival analysis. Horizontal coordinates indicate survival time, and vertical coordinates indicate survival rate. Red line: high TMB group + high m⁶A score; blue line: high TMB group + low m⁶A score; purple line: low TMB group + high m⁶A score; green line: low TMB group + low m⁶A score.

inflammatory features. m⁶A score is a reliable tool that can be used to assess m⁶A-TME. TMB alone is not a good predictor for the effect of immune checkpoint inhibitors, and the combined m⁶A score can improve the ability of TMB to predict immunotherapy [25–27]. Further studies revealed that higher PD-L1 expression in low m⁶A received better immunotherapy, suggesting that the lymphoid infiltrating cells in the tumor microenvironment can enhance the efficacy of immune check blocking therapy, while stromal cells can exert an anti-immune check blocker effect.

## Conclusions

In clinical practice, the m⁶A score can be used to comprehensively evaluate the m⁶A methyla-tion modified regulators and their corresponding TME cell infiltration characteristics in indi-vidual patients, so as to determine the immune phenotype of tumors and guide clinical practice more effectively. In addition, the m⁶A score can be used to predict the clinical

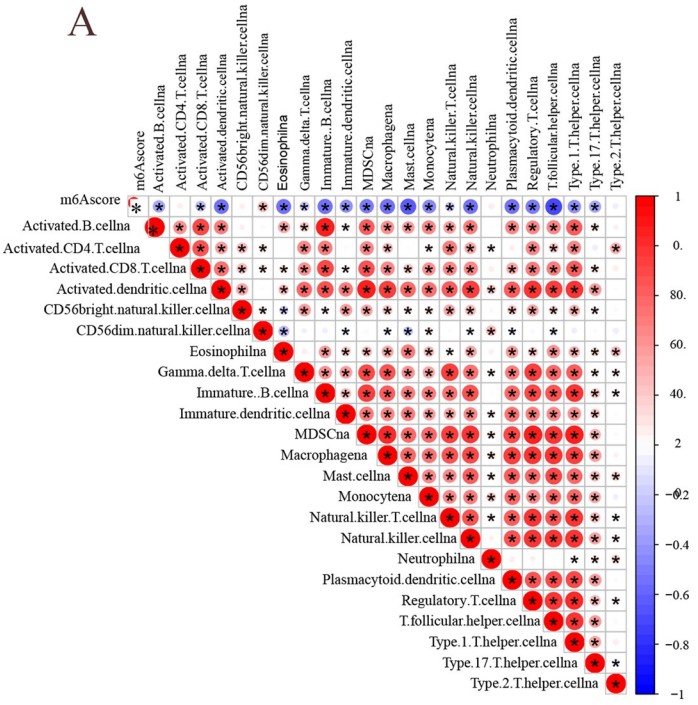

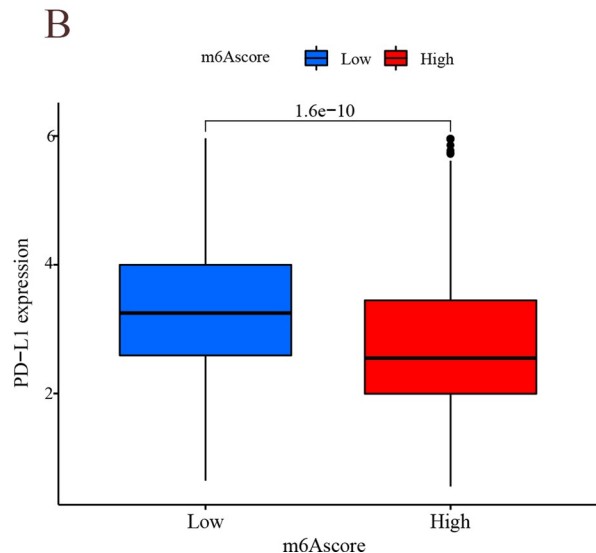

**Fig 15. Analysis of immune˚immune correlation.** (A) Immunocorrelation analysis. Horizontal and vertical coordinates indicate immune infiltrated cells and m⁶A score, while the intersecting circles indicate the correlation between them. Besides, red indicates positive correlation, and blue indicates a negative correlation. The darker the color, the larger the circle, and the closer the correlation. (B) The horizontal coordinate represents the m⁶A subgroup and the vertical coordinate represents PD-L1 expression.

response to adjuvant immunotherapy. More importantly, this research provides some new insights into tumor immunotherapy by targeting m⁶A regulators for altering the m⁶A modified phenotype to convert cold tumors into hot tumors, which may provide a novel idea for the development of new drug combination strategies in the future.

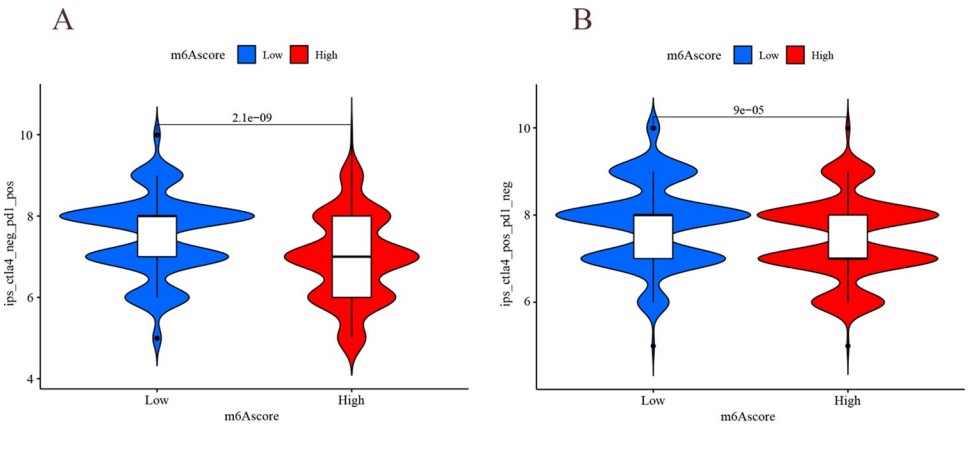

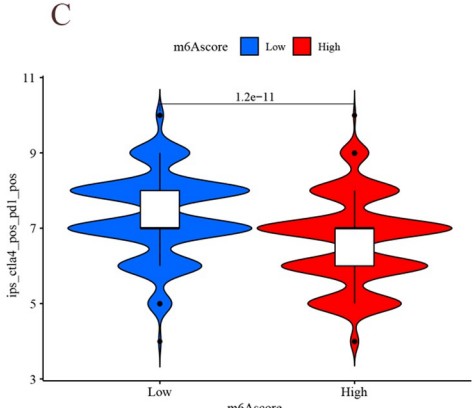

**Fig 16. Analysis of immune˚checkpoint molecules.** (A) Horizontal coordinate indicates m⁶A score, and vertical coordinate indicates the effect of PD-L1 treatment. (B) Horizontal coordinates indicate m⁶A score, and vertical coordinates indicate the effect of CTLA4 treatment. (C) Horizontal coordinates indicate m⁶A score, and vertical coordinates indicate CTLA4 combined with the effect of PD-L1 treatment.

## Supporting information

**S1 Fig. Raw data acquisition process from database.**
(TIF)

**S2 Fig. The overall step-by-step process of this work.**
(TIF)

**S1 Table. Prognostic analysis of m⁶A regulators using a univariate Cox regression model.**
(DOCX)

**S2 Table. The changes of m⁶A cluster.**
(DOCX)

**S3 Table. Prognostic analysis of Intersecting genes using a univariate Cox regression model.**
(DOCX)

**S4 Table. The changes of gene cluster.**
(DOCX)

**S5 Table. The changes of m⁶A score.**
(DOCX)

**S1 Data.**
(ZIP)

## Acknowledgments

Not applicable. Competing interests. The authors declare that they have no competing interests, and all authors should confirm its accuracy.

## Author Contributions

**Data curation:** Tianpeng Huang.

**Formal analysis:** Tianpeng Huang.

**Methodology:** Tianpeng Huang.

**Resources:** Tianpeng Huang.

**Validation:** Wei Ye.

**Writing – original draft:** Wei Ye.

**Writing – review & editing:** Wei Ye.

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
