## [Editor Report · Decision Letter 0]

21 Jul 2021

PONE-D-21-22417

Correlation analysis of m6A-modified regulators with immune microenvironment infiltrating cells in lung adenocarcinoma

PLOS ONE

Dear Dr. Huang,

Thank you for submitting your manuscript to PLOS ONE. After careful consideration, we feel that it has merit but does not fully meet PLOS ONE’s publication criteria as it currently stands. Therefore, we invite you to submit a revised version of the manuscript that addresses the points raised during the review process.

Methods needs clarification. Providing step by step details of methods is required. Data needs to be provided.

We look forward to receiving your revised manuscript.

Kind regards,

Esmaeil Ebrahimie, Ph.D.

Academic Editor

PLOS ONE

Journal Requirements:

"No" 

"No"

Additional Editor Comments:

Dear Authors,

The manuscript I based on the data gathered from TCGA database and GEO databased. However, little information is provided in the Methods regarding how the data is searched, and downloaded.

Also, the methods and packages need detailed explanations and clarifications.

Major comments

Major comment 1.

There is a necessity that the authors provide a detailed step by step visualized guidelines or video recording determines:

1-1-How the data is searched (search key words) in TCGA and GEO

1-2-How the data is downloaded

The mentioned guideline can be uploaded as Supplementary file.

Major comment 2.

Detailed references of R packages used for GSVA analysis and ssGSEA analysis as well as

differential expression using R limma package need to be provided.

R codes needs to be attached.

I do suggest to provide step by step visualized guidelines or video recording

Major comment 3.

All downloaded raw data need to be shared.

Major comment 4.

A detailed step by step visualized guidelines or video recording of performing Survival analysis and Survival curves needs to be provided.

Major comment 5.

Data of two immunotherapy cohorts needs to be provided, including raw data as well as cohort statistics.

Minor comments

Minor comment 1.

Quality of figures need to be improved.

Minor comment 2.

A flowchart describing all steps of analysis needs to be provided.

Examples can be provided here:

https://www.frontiersin.org/articles/10.3389/fpls.2018.01550/full

https://www.nature.com/articles/s41598-019-45661-7

Minor comment 3.

English of manuscript needs to be improved. There are typs in manuscript. I suggest using English Editing service.

Please note that the work is a bioinformatic and statistics study. I am not able to send the work to review until the Methos is documented, clarified, and become reproducible.
---

## [Author Response · Author response to Decision Letter 0]

6 Aug 2021

1.There is a necessity that the authors provide a detailed step by step visualized guidelines or video recording determines:

1-1-How the data is searched (search key words) in TCGA and GEO

1-2-How the data is downloaded

The mentioned guideline can be uploaded as Supplementary file.

ANSWER:The description of the data collection is supplemented in the Methods and Materials section. Further, visual graphics were made for representation and uploaded as an Supplementary file（Figure-S1）.

2.Detailed references of R packages used for GSVA analysis and ssGSEA analysis as well as differential expression using R limma package need to be provided.

R codes needs to be attached.

I do suggest to provide step by step visualized guidelines or video recording

ANSWER:Detailed references for GSVA analysis and ssGSEA analysis and R packages for differential expression using the limma package have been described in the Methods and Materials section and are accompanied by the R code.

3.All downloaded raw data need to be shared.

ANSWER:All materials have been attached and uploaded as follows：

1. TCGA-LUAD (annotated document)

2.GEO dataset (platform files and matrix files)

3.TCIA dataset (TCIA-ClinicalData)

4. ssGSEA subsidiary files (immune.gmt)

5. GSVA files (c5.go.v7.4.symbols)

4.A detailed step by step visualized guidelines or video recording of performing Survival analysis and Survival curves needs to be provided.

ANSWER:Details of the survival analysis and survival curve are described in detail in the methods and materials section.

5.Data of two immunotherapy cohorts needs to be provided, including raw data as well as cohort statistics.

ANSWER：Raw data has been uploaded via attachments as well as statistical data as Supplementary file.

6.Quality of figures need to be improved.

ANSWER:Figure has been appropriately modified.

7.A flowchart describing all steps of analysis needs to be provided.

ANSWER:We have added the flow chart and uploaded it as an Supplementary file（Figure-S2）.

---

## [Decision Letter · Decision Letter 1]

30 Dec 2021

PONE-D-21-22417R1Correlation analysis of m6A-modified regulators with immune microenvironment infiltrating cells in lung adenocarcinomaPLOS ONE

Dear Dr. Huang,

Thank you for submitting your manuscript to PLOS ONE. After careful consideration, we feel that it has merit but does not fully meet PLOS ONE’s publication criteria as it currently stands. Therefore, we invite you to submit a revised version of the manuscript that addresses the points raised during the review process.

The manuscript has been evaluated by the reveiwers and requires some minor revisions including typos and table formatting. Kindly pay attention to these comments and resubmit the manuscript with all requested changes. 

We look forward to receiving your revised manuscript.

Kind regards,

Afsheen Raza, PhD

Academic Editor

PLOS ONE

Journal Requirements:

Reviewers' comments:

Reviewer's Responses to Questions

**Comments to the Author**

1. If the authors have adequately addressed your comments raised in a previous round of review and you feel that this manuscript is now acceptable for publication, you may indicate that here to bypass the “Comments to the Author” section, enter your conflict of interest statement in the “Confidential to Editor” section, and submit your "Accept" recommendation.

Reviewer #1: All comments have been addressed

Reviewer #2: All comments have been addressed

2. Is the manuscript technically sound, and do the data support the conclusions?

Reviewer #1: Yes

Reviewer #2: Yes

3. Has the statistical analysis been performed appropriately and rigorously? 

Reviewer #1: Yes

Reviewer #2: Yes

4. Have the authors made all data underlying the findings in their manuscript fully available?

Reviewer #1: Yes

Reviewer #2: Yes

5. Is the manuscript presented in an intelligible fashion and written in standard English?

Reviewer #1: Yes

Reviewer #2: No

6. Review Comments to the Author

Reviewer #1: (No Response)

Reviewer #2: The authors had made many typos and misused punctuations in the manuscript. The corresponding content of the tables and other supplementary informations should be marked in the main text.

7. PLOS authors have the option to publish the peer review history of their article (what does this mean?). If published, this will include your full peer review and any attached files.

Reviewer #1: No

Reviewer #2: No

---

## [Author Response · Author response to Decision Letter 1]

4 Jan 2022

Dear Editor

Many thanks for the suggestions on the manuscript that can improve the quality of the manuscript. In response to the returned suggestions, the following corrections were made, some of which are marked in red in the manuscript, with the following descriptions：

1.The manuscript has been evaluated by the reveiwers and requires some minor revisions including typos and table formatting. Kindly pay attention to these comments and resubmit the manuscript with all requested changes. 

Re: Correction of the manuscript for punctuation and grammar. The table format was corrected according to the journal's requirements and some adjustments were made to make the manuscript more logical.

If there are still any questions, feel free to contact us and your comments are welcome!

 Sincerethanks 

TianPeng Huang

---

## [Editor Report · Decision Letter 2]

10 Feb 2022

Correlation A nalysis of m 6 A-modified R egulators with I mmune M icroenvironment I nfiltrating C ells in L ung A denocarcinoma

PONE-D-21-22417R2

Dear Dr. Huang,

We’re pleased to inform you that your manuscript has been judged scientifically suitable for publication and will be formally accepted for publication once it meets all outstanding technical requirements.

Kind regards,

Afsheen Raza, PhD

Academic Editor

PLOS ONE
---

## [Editor Report · Acceptance letter]

14 Feb 2022

PONE-D-21-22417R2 

Correlation Analysis of m^6^A-modified Regulators with Immune Microenvironment Infiltrating Cells in Lung Adenocarcinoma 

Dear Dr. Huang:

I'm pleased to inform you that your manuscript has been deemed suitable for publication in PLOS ONE. Congratulations! Your manuscript is now with our production department. 

Kind regards, 

on behalf of

Dr. Afsheen Raza 

Academic Editor

PLOS ONE